# Aging in zebrafish is associated with reduced locomotor activity and strain dependent changes in bottom dwelling and thigmotaxis

**Jacob Hudock**[ID]**, Justin W. Kenney**[ID]*

Department of Biological Sciences, Wayne State University, Detroit, MI, United States of America

* gl4448@wayne.edu

**Data Availability Statement:** We have uploaded a CSV file containing all the underlying data as a supplemental file.

## Abstract

Aging is associated with a wide range of physiological and behavioral changes in many species. Zebrafish, like humans, rodents, and birds, exhibits gradual senescence, and thus may be a useful model organism for identifying evolutionarily conserved mechanisms related to aging. Here, we compared behavior in the novel tank test of young (6-month-old) and middle aged (12-month-old) zebrafish from two strains (TL and TU) and both sexes. We find that this modest age difference results in a reduction in locomotor activity in male fish. We also found that background strain modulated the effects of age on predator avoidance behaviors related to anxiety: older female TL fish increased bottom dwelling whereas older male TU fish decreased thigmotaxis. Although there were no consistent effects of age on either short-term (within session) or long-term (next day) habituation to the novel tank, strain affected the habituation response. TL fish tended to increase their distance from the bottom of the tank whereas TU fish had no changes in bottom distance but instead tended to increase thigmotaxis. Our findings support the use of zebrafish for the study of how age affects locomotion and how genetics interacts with age and sex to alter exploratory and emotional behaviors in response to novelty.

## Introduction

The challenge of understanding the biological and behavioral effects of age is of central concern in medicine and biology and is facilitated by the comparative study of aging across species [1, 2]. Zebrafish, like humans, display gradual senescence, despite their indeterminate growth [3]. With advancing age, zebrafish have reduced regenerative capacity, accumulation of DNA damage, shortening of telomeres, and cardiovascular impairment [4–6]. This is in contrast to other species with indeterminate growth, like many reptiles and sea urchins, that tend to age very slowly [7, 8]. The shared consequences of aging in humans and zebrafish, combined with the low cost of housing zebrafish for extended periods, suggests that zebrafish could serve as a valuable model organism in the study of aging.

In humans, many behaviors vary across the lifespan, such as emotional regulation and anxiety [9–11]. The trajectory of these behaviors is shaped by the complex interplay of

**Funding:** This work was funded by the NIGMS, R35GM142566. The funders had no role in the study design, data collection and analysis, decision to publish, or preparation of the manuscript.

**Competing interests:** The authors have declared that no competing interests exist.

environmental factors with genetics and sex [11, 12]. Untangling the web of interactions between age, genetics, and sex, is facilitated by use of model organisms for which many molecular genetic tools are available, like zebrafish. Although there has been a large increase in the study of zebrafish behavior in the past twenty years [13, 14], only a handful of reports have explored age-associated behavioral alterations. These studies have found that older fish have changes in circadian rhythms, decreased associative learning, reduced locomotion, and elevated bottom dwelling [5, 15–21]. However, it is unknown if genetics and sex interact with age to affect commonly studied behaviors related to emotional regulation and exploration in zebrafish.

One of the most widely used behavioral tasks to study exploratory and emotional behavior in zebrafish is the novel tank test (NTT). In this test, animals are placed into a tank with which they have no prior experience. Spending time near the bottom or periphery of the tank are typically interpreted as fearful predator avoidance behaviors related to anxiety [22–26]. Non-associative memory can also be assessed by measuring how behavior habituates within session (short-term memory) or between daily sessions (long-term memory). To determine whether age affects behavior in the NTT, and if the effects of age are modulated by sex and genetics, we examined behavior in young (6 months old) and middle aged (12 months old) fish from two strains (TL and TU) and both sexes. We find that this modest difference in age is sufficient to alter several behaviors such as locomotion and predator avoidance.

## Results

### The effects of age on bottom dwelling and thigmotaxis

We measured behavior during exploration of a novel tank in young (6 mpf) and old (12 mpf) zebrafish from both sexes and two strains, TLs and TUs. Distance from the bottom of the tank is considered a predator avoidance response that is one of the most widely studied behaviors in zebrafish [24, 27]. This behavior has also been interpreted as 'anxiety-like', where spending more time near the bottom of the tank indicates greater anxiety [26]. In TL fish, there was a medium sized effect of age on distance from the bottom of the tank (Fig 1A, top; P = 0.035, $\eta^2$ = 0.08) where older fish had greater anxiety-like behavior, spending more time near the bottom of the tank. There was no effect of sex (P = 0.36) or an interaction (P = 0.21). Post-hoc tests within sex found the effect of age on bottom dwelling to be present in females (P = 0.033, d = 0.90), but not males (P = 0.51). In TU fish, age did not affect bottom dwelling (Fig 1B, bottom, P = 0.14) and there was no effect of sex (P = 0.23) or an interaction (P = 0.30).

Thigmotaxis (i.e. time spent near the periphery) of a tank is also commonly interpreted as a predator avoidance response in zebrafish, although this interpretation has recently come under question (see discussion section). In TL fish (Fig 1B, top), there were no effects of age (P = 0.68), sex (P = 0.96) or an interaction (P = 0.23) on center distance. However, in TU fish (Fig 1B, bottom), there was a medium-sized effect of age (P = 0.013, $\eta^2$ = 0.12) where older fish spent more time in the center of the tank than younger fish. There was no effect of sex (P = 0.92) or an interaction (P = 0.12). Post-hoc tests within sex identified that the effect of age on center distance was limited to males, where older males spent more time closer to the center of the tank than young males (P = 0.045, d = 1.13); there was no difference in females (P = 0.44). Taken together with the findings on bottom distance, the effects of age on predator avoidance/anxiety-like behaviors depends on the strain of the fish: older TL fish, especially females, spend more time near the bottom of the tank whereas older TU fish, particularly males, spend more time in the center of the tank.

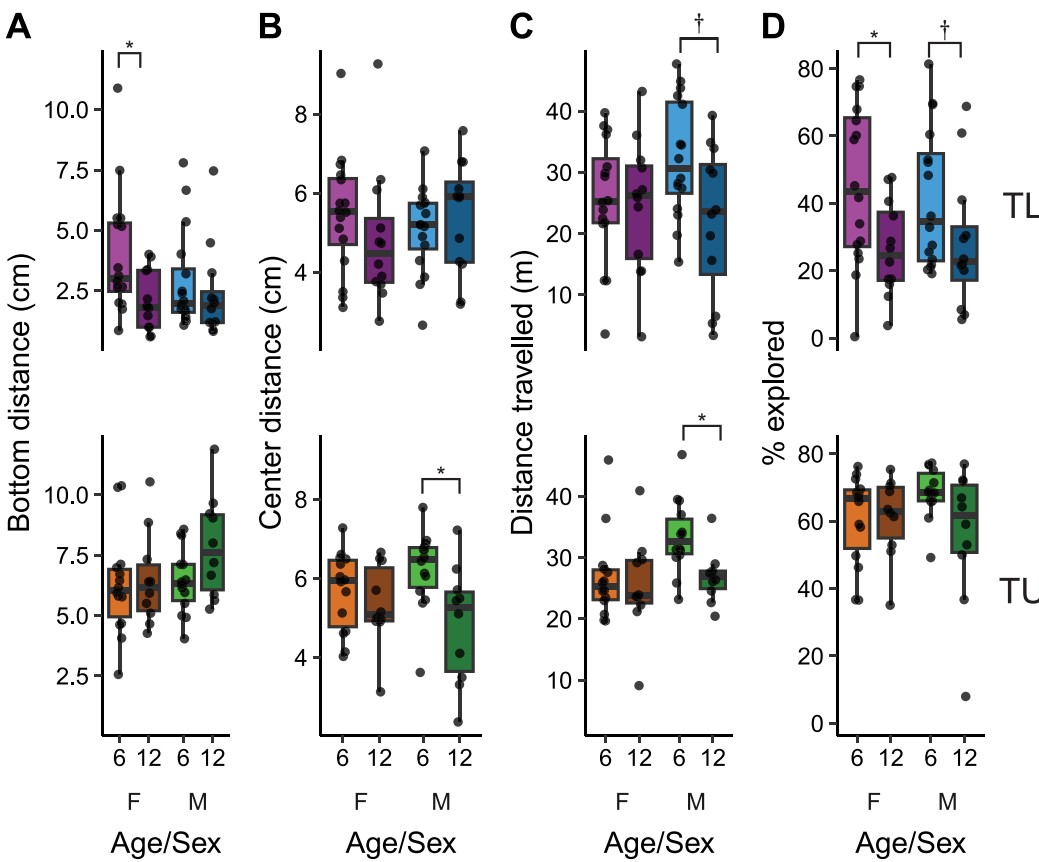

**Fig 1.** Behavior during an initial exposure to a novel tank in young (6 mpf) and old (12 mpf) TL (top) and TU (bottom) fish of both sexes. Behaviors assessed were (A) bottom distance, (B) center distance, (C) distance travelled and (D) percent of the tank explored. Boxplot center line is the median, hinges are the interquartile range, and whiskers are hinges ± 1.5 times the interquartile range. *P < 0.05, †P < 0.10 from FDR corrected post-hoc t-tests within sex.

## The effects of age on locomotion and exploration

Aging in zebrafish has been associated with a decrease in activity levels [17]. To determine if this also occurred during the NTT, we measured how far fish travelled during their exposure to the novel tank (Fig 1C). In TL fish, there was a strong trend towards a medium-sized effect of older fish swimming less than younger fish (P = 0.057, $\eta^2$ = 0.064) with no effect of sex (P = 0.32) or an interaction (P = 0.12). Post-hoc tests indicated a strong trend towards reduced locomotion due to age in males (P = 0.057, d = 0.93), but not females (P = 0.78). In TU fish, there was a medium-sized effect of age (P = 0.044, $\eta^2$ = 0.077) with older fish also swimming less than younger fish. There was also an effect of sex in TUs (P = 0.022, $\eta^2$ = 0.10) where males swam further than females; there was no interaction between age and sex (P = 0.17). Post-hoc tests within sex found that young male TU fish swam further than their older counterparts (P = 0.009, d = 1.24) with no effect in females (P = 0.68). Thus, older fish, particularly males, swim less than their younger counterparts, an effect that is largely independent of strain.

To determine if age influences how much of the novel tank fish choose to explore, we measured the percentage of the tank visited by fish (Fig 1D). In TL fish, there was a strong effect of older fish exploring less of the tank than younger fish (P = 0.0052, $\eta^2$ = 0.14). There was no effect of sex (P = 0.79) or an interaction (P = 0.61). Post-hoc tests indicated that the effect of

age is more prominent in female fish (P = 0.029, d = 0.93) than male animals (P = 0.10, d = 0.64). In TU fish, there was no effect of age (P = 0.14) or sex (P = 0.54) on percent explored. There was a trend towards a small interaction (P = 0.10, $\eta^2$ = 0.055), although post-hoc tests did not uncover any differences. These findings suggest that background strain strongly modulates the influence of age on the willingness to explore a new environment, having a bigger impact on TLs than TUs. This is unlikely to be due entirely to less locomotor activity in older fish because older TU fish swam less than their younger counterparts with no impact on percent explored.

## The effects of age on short-term habituation

When exposed to a novel environment many animals, including fish, habituate such that their behaviors change within session [25, 28, 29]. To determine if habituation occurred during exposure to the novel tank, and whether this habituation was affected by age, we looked at how predator avoidance behaviors and locomotion changed between the first two and last two minutes in the novel tank (Fig 2). For changes in bottom distance in TL fish (Fig 2B, top), we found no effects of age (P = 0.40), sex (P = 0.66) or an interaction (P = 0.86). When compared to zero (i.e., no change), we did find a strong effect in older male animals, which increased their bottom distance over time (P = 0.0094, d = 1.1), while younger males and older females had trends towards medium sized increases in bottom distance (P's = 0.063 & 0.065, d's = 0.59 & 0.64, respectively) and young females did not change (P = 0.16). The lack of effect in young TL females may reflect the fact that they had high bottom distance from the beginning of the trial (Fig 2A). In TU fish, there were medium size effects of age (P = 0.030, $\eta^2$ = 0.094) and sex (P = 0.045, $\eta^2$ = 0.079) with no interaction (P = 0.59): older fish increased bottom dwelling over time (Fig 2B, bottom). When comparing the difference to zero, however, the results were less clear. Older female fish had a trend towards a strong effect of decreasing bottom distance (P = 0.066, d = 0.93) whereas young females (P = 0.84), young males (P = 0.27), and older males (P = 0.84) did not change their bottom dwelling behavior over time.

For change in distance from the center of the tank over time, TL fish did not appear to habituate whereas TU fish increased center distance over time (Fig 2C). In examining the change in center distance between the first two and last two minutes, in TL fish (Fig 2D, top) there were no effects of age (P = 0.40), sex (P = 0.66), or an interaction (P = 0.42). Furthermore, none of the TL fish showed changes in center distance over time (P's > 0.84). Likewise, in TU fish (Fig 2D, bottom), there were also no effects of age (P = 0.44), sex (P = 0.50) or an interaction (P = 0.99). However, in contrast to the TL fish, in all groups of the TU fish there were large effects of increased center distance over time (P's < 0.01, d's > 1).

Locomotor activity has also been found to change during exposure to a novel environment [25, 30]. An initial look at the minute-by-minute data (Fig 2E) suggests that TL and TU fish differ in that TL fish decreased distance travelled over time whereas TU fish increased distance travelled. In TL fish, there were no effects of age (P = 0.29), sex (P = 0.52) or an interaction (P = 0.44) on the change in distance travelled over time (Fig 2F, top). Only older male fish significantly decreased over time (P = 0.024, d = 0.98; all other P's > 0.15). In TU fish, there was a medium sized effect of age (P = 0.039, $\eta^2$ = 0.092) on the change in distance travelled (Fig 2F, bottom) such that older fish increased their distance travelled between the beginning and end of the session more than younger fish. There was no effect of sex (P = 0.49) or an interaction (P = 0.73). When comparing distance travelled in the first two minutes to the last two minutes, there were strong effects of increased locomotor activity over time in young female (P = 0.012, d = 0.87), older female (P = 0.012, d = 1.14) and older male fish (P = 0.027, d = 0.89). There was no change in young male fish (P = 0.20).

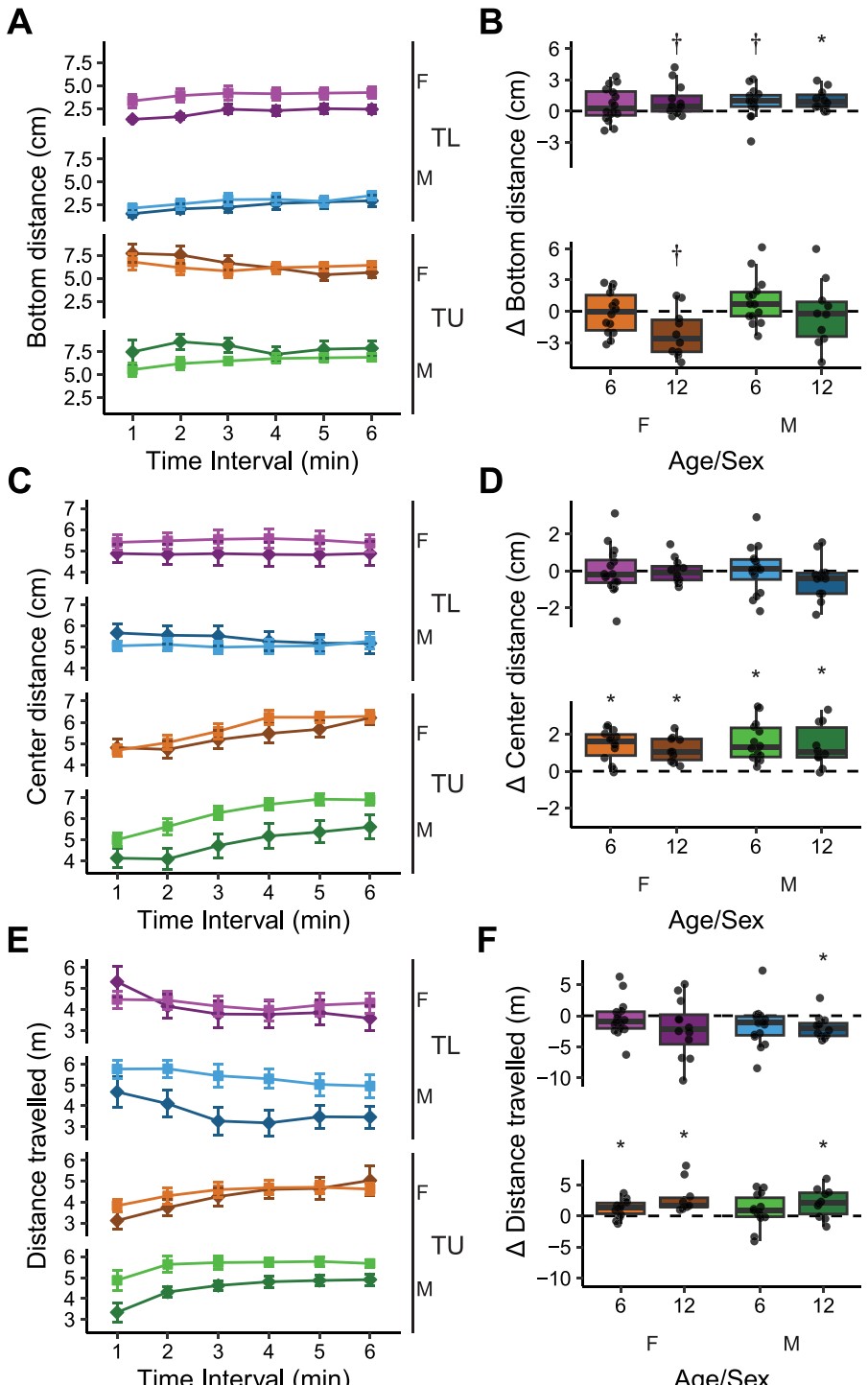

**Fig 2. Behavior across time during exploration of a novel tank in young (6 mpf) and old (12 mpf) TL and TU fish of both sexes.** Behaviors assessed were (A) minute-by-minute bottom distance, (B) difference in bottom distance during the last two minutes compared to the first two minutes, (C) minute-by-minute center distance, (D) difference in center distance during the last two minutes compared to the first two minutes, (E) minute-by-minute distance travelled, (F) difference in distance travelled during the last two minutes compared to the first two minutes. Boxplot center line is the median, hinges are the interquartile range, and whiskers are hinges ± 1.5 times the interquartile range. *P < 0.05, †P < 0.10 compared to zero using one-sample t-tests with FDR corrections.

## The effect of age on bottom dwelling and thigmotaxis during a second exposure to the novel tank

We next sought to determine if the effects of age we saw during the initial exposure to the novel tank persisted during a second exposure 24 hours later. For bottom distance in TL fish (Fig 3A, top), there was no longer a main effect of age (P = 0.20). There was also no effect of sex (P = 0.29), but there was a strong trend towards a medium-sized interaction between age and sex (P = 0.055, $\eta^2$ = 0.069) where older female fish had lower bottom distance than their younger counterparts with the opposite trend in males. Following up the interaction with post-hoc tests within sex indicated a trend towards a large effect in older versus younger TL females (P = 0.088, d = 0.80) and no effect in males (P = 0.58). Thus, the higher anxiety-like behavior in older females weakly persisted from day 1 to day 2 in TL fish. In TU fish, there were no effects of age (P = 0.51) or sex (P = 0.46) on bottom distance on the second day, although there was a trend towards a medium-sized interaction between age and sex (P = 0.081, $\eta^2$ = 0.067; Fig 3A, bottom) mirroring what was observed in TL fish. However, post-hoc tests within sex found no differences between young and old fish in either females (P = 0.43) or males (P = 0.28).

In TL fish on day 2, center distance (Fig 3B, top) was not influenced by age (P = 0.92), sex (P = 0.90) nor was there an interaction between age and sex (P = 0.75). In TUs, there was no longer an effect of age on center distance on day 2 (P = 0.13), nor was there an effect of sex (P = 0.76) or an interaction (P = 0.33).

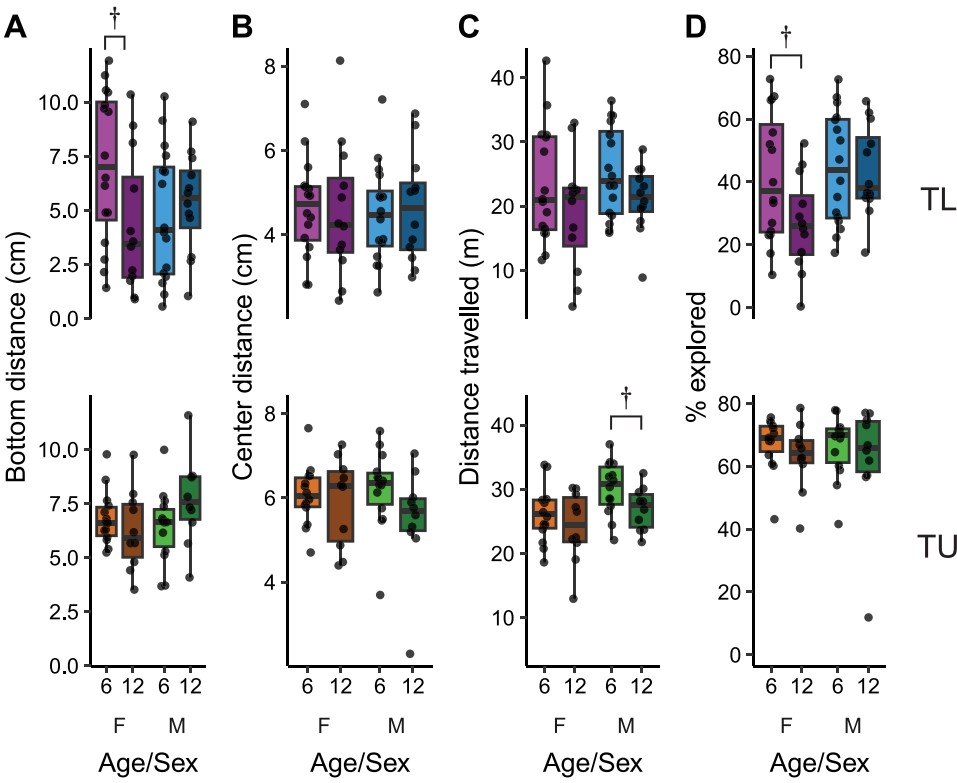

**Fig 3.** Behavior during a second exposure to the tank in young (6 mpf) and old (12 mpf) TL (top) and TU (bottom) fish of both sexes. Behaviors assessed were (A) bottom distance, (B) center distance, (C) distance travelled and (D) percent of the tank explored. Boxplot center line is the median, hinges are the interquartile range, and whiskers are hinges ± 1.5 times the interquartile range. †P < 0.10 from FDR corrected post-hoc t-tests within sex.

## The effects of age on locomotion and exploration during a second exposure to the tank

In TL fish, as on the first day, there was a trend towards a medium-sized effect of age on distance travelled (P = 0.054, $\eta^2$ = 0.068) where older fish swam less than younger fish. There was no effect of sex (P = 0.34) or an interaction (P = 1; Fig 3C, top), and post-hoc tests within sex were not significant in either females (P = 0.24) or males (P = 0.17). The effects of age on locomotor activity in TU fish also persisted to the second day (Fig 3C, bottom): there was a medium-sized effect of age (P = 0.049, $\eta^2$ = 0.074) such that older fish swam less than younger fish. There was also a large effect of sex (P = 0.011, $\eta^2$ = 0.14) with male fish swimming further than female fish. There was no interaction between age and sex (P = 0.64). Post-hoc tests within sex found no effect in female TU fish (P = 0.36), and only a trend in males (P = 0.096, d = 0.84). Overall, the main effects of age on locomotor activity observed on day 2 were largely consistent with what was seen on day 1, suggesting these behavioral differences are particularly stable.

The exploration of an environment may also change upon repeated exposure due to increased familiarity. During the second exposure to the tank, in TL fish, there were trends towards small effects of age (P = 0.10, $\eta^2$ = 0.046) and sex (P = 0.068, $\eta^2$ = 0.058) with no interaction (P = 0.16; Fig 3D). Post-hoc tests found that only young female TLs had a trend towards greater exploration than their older counterparts (P = 0.076, d = 0.80) with no effect in males (P = 0.87). On the second day in TU fish, there were no effects of age (P = 0.24), sex (P = 0.81) or an interaction (P = 0.98). Taken together, as we saw on day 1, the influence of age on exploration of the novel tank was more prominent in TLs, albeit with smaller effect sizes than observed on day 1.

## The effects of age on long-term habituation

To determine if fish exhibited long-term habituation, we calculated how their behavior changed on day 2 compared to day 1 (Fig 4). In TL fish, we found that all groups of animals increased their bottom distance on the second day (P's ≤ 0.026, d's: 0.74–1.23; Fig 4A, top). There were no effects of age (P = 0.88), sex (P = 0.64) or an interaction (P = 0.24) on the change in distance from bottom. In contrast, bottom distance in TU fish did not habituate on the second day (P's > 0.47; Fig 4A, bottom), and there were no effects of age (P = 0.22), sex (P = 0.43) or an interaction (P = 0.46). This lack of an effect in TUs may represent a ceiling effect as they had higher baseline bottom distance on day 1 (Fig 1A, bottom).

For long-term habituation of center distance, all groups of TL fish had medians below zero, decreasing their center distance (Fig 4B, top), although only males of both ages approached statistical significance (P's = 0.084; d's = 0.64 and 0.66 for young and old males, respectively; females: P's ≥ 0.13). There were no effects of age (P = 0.59), sex (P = 0.86) or an interaction (P = 0.33) on the change in center distance in TL fish. In contrast, most TU fish increased their center distance on day 2 (Fig 4B, bottom). There were trends towards medium-sized effects in females (P's = 0.070, d = 0.63 & 0.80 for young and older females, respectively) and older males (P = 0.084, d = 0.67), but not younger males (P = 0.94). There was also a trend towards a medium-sized effect of age on the change in center distance (P = 0.057, $\eta^2$ = 0.73) such that older animals had a greater increase in center distance than younger animals. There were no effects of sex (P = 0.43) or an interaction (P = 0.28). Post-hoc tests within sex did not uncover any significant differences (female: P = 0.54, male: P = 0.17).

The median locomotor activity in TL fish also trended towards a decrease on the second day (Fig 4C, top), although there was only a significant decrease in young males (P = 0.031, d = 0.77) and a trend in older females (P = 0.058, d = 0.72) with no effect in younger females

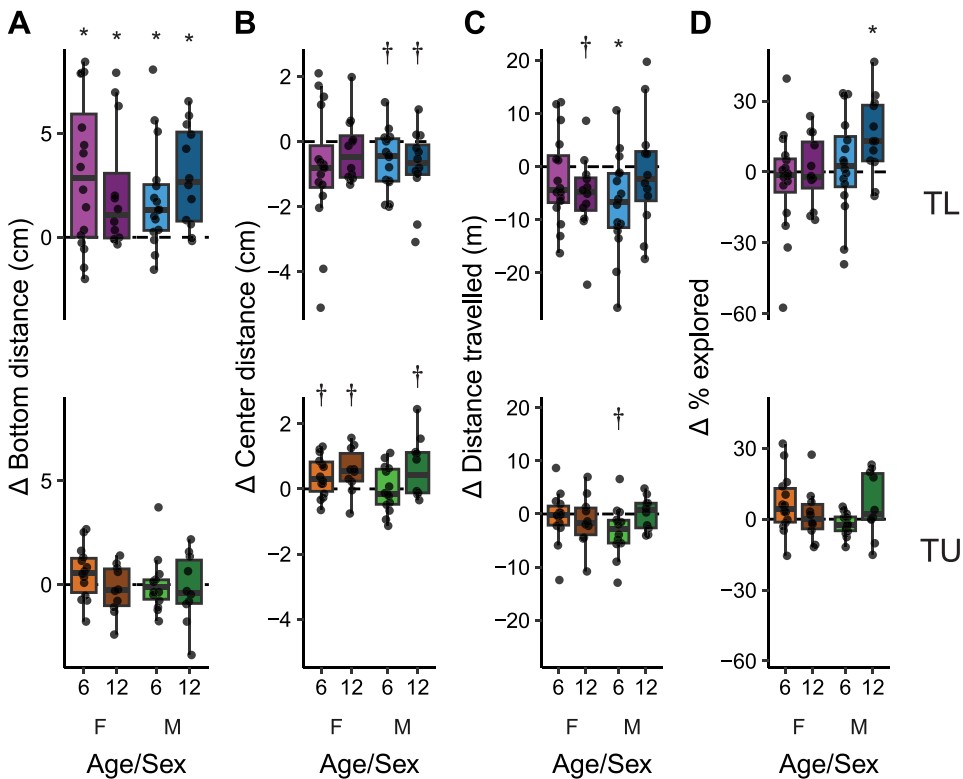

**Fig 4.** Change in behavior between the first and second days of exposure to a novel tank in young (6 mpf) and old (12 mpf) TL (top) and TU (bottom) fish of both sexes. Behaviors assessed were changes in (A) bottom distance, (B) center distance, (C) distance travelled and (D) percent of the tank explored. Boxplot center line is the median, hinges are the interquartile range, and whiskers are hinges ± 1.5 times the interquartile range. *P < 0.05, †P < 0.10 from one-sample t-tests compared to zero with FDR corrections.

(P = 0.38) or older males (P = 0.74). When comparing the groups to each other, there were no main effects of age (P = 0.55) or sex (P = 0.73), but there was a trend towards a medium sized interaction of age and sex (P = 0.070, $\eta^2$ = 0.062) where, compared to their younger counterparts, older female fish decreased, and older male fish increased, locomotor activity from day 1 to day 2. However, follow-up post-hoc tests within sex found no differences in females (P = 0.32) or males (P = 0.28). Like the TLs, young TU males had a trend towards a decrease in locomotor activity on the second day (P = 0.085, d = 0.70) with no change in any other groups (P's ≥ 0.86; Fig 4C, bottom). The change in locomotion in TUs was not affected by age (P = 0.31) or sex (P = 0.63), and there was no interaction (P = 0.13).

The percentage of the tank explored on the second day changed little in both TL and TU fish (Fig 4D; P's > 0.24) except for older male TL's (P = 0.042, d = 0.89). In TL fish there was a trend towards a medium-sized effect of sex on the change in percent explored (P = 0.055, $\eta^2$ = 0.069) where males, irrespective of age, increased their percent explored compared to females. There were no effects of age (P = 0.12) or an interaction (P = 0.45), and post-hoc tests found no differences within each sex (female: P = 0.55, male: P = 0.21). In TU fish, there was no effect of age (P = 0.58) or sex (P = 0.30). There was a medium-sized interaction (P = 0.047, $\eta^2$ = 0.87) where young female and older male TUs increased their exploration compared to older females and younger males, however, follow-up post-hoc tests within sex found no differences in females (P = 0.35) or males (P = 0.19).

## Discussion

The behavior of an organism arises through an interaction between environmental conditions and biological factors like genetics, sex, and age. Understanding how these factors interact to influence behavior represents a major challenge in behavioral neurobiology that has important implications for improving human health. Here, we found behavioral changes during exploration of a novel tank in zebrafish that differed in age by only six-months (6 mpf versus 12 mpf). The most consistent behavioral change was a decrease in locomotor activity in older male animals. The effects of age on predator avoidance behaviors related to anxiety were more complex as they were sensitive to genetic background and prior exposure to the tank. There were no clear effects of aging on either short-term or long-term habituation.

We found that age-related changes in the anxiety-related exploratory behavior of zebrafish are modulated by background strain: older TL fish had increased bottom dwelling (i.e., elevated anxiety-like behavior), whereas older TU fish had a decrease in thigmotaxis (i.e., decreased anxiety-like behavior, but see discussion below on challenges for interpreting thigmotaxis in zebrafish). Our findings with bottom dwelling are consistent with prior work in zebrafish where older animals spent more time near the bottom of a novel tank [15, 16, 19]. The complexity of how genetics interacts with age in anxiety-related behaviors has also been found in other model organisms, such as mice, where age is associated with both increases and decreases in anxiety, depending on the specific measure or test, [31], and genetic background [32–34]. These findings in animals mirror some of the complexity observed in humans, where genetic contributions to fear and anxiety vary across anxiety phenotypes [35]. Thus, the present study suggests that zebrafish can be used to model how genetics influences the aging process.

We found that older male TU fish had a decrease in thigmotaxis (i.e., spent more time closer to the center of the tank). Thigmotaxis in zebrafish has typically been interpreted as predator avoidance and thus anxiety-related based on its similarity to behavior in rodents [22, 24, 26]. However, recent findings call this interpretation into question. For example, bottom dwelling is better established as a predator avoidance behavior related to anxiety [24, 26, 27, 36], and when measured concurrently with thigmotaxis, these two parameters do not consistently correlate with each other [37]. Furthermore, several studies have found that animals with high levels of freezing or immobility, another well-defined index of fear and anxiety [26, 38], spend more time in the center of the tank than the periphery [39, 40]. We, and others, have also found thigmotaxis to increase over time in a novel environment in some fish strains (Fig 2C) [22, 37, 41], which is the opposite of what would be expected as animals habituate. Finally, the anxiolytic and anxiogenic effects of drugs often have the expected effects on bottom dwelling without altering thigmotaxis [42, 43]. Thus, the interpretation of thigmotaxis in zebrafish is not straightforward. The present study suggests the interesting possibility that whether thigmotaxis is indicative of predator-avoidance and anxiety may be related to genetic background: TU fish display several changes in thigmotaxis (e.g., short-term habituation [Fig 2C], long-term habituation [Fig 4B], and the effects of age [Fig 1B]) whereas effects in TL fish are largely confined to bottom dwelling. When TL fish do have changes in thigmotaxis (i.e., long-term habituation, Fig 4B) it is reduced instead of increased like it is in TU fish. This idea that genetic background may influence the interpretation of thigmotaxis in zebrafish warrants further study.

The most consistent effect of age we observed was decreased locomotion in older fish, particularly in males. This effect was present in both strains and both days that animals were placed in the tank and is in line with prior work [5, 16, 17, 20]. Indeed, age related decreases in locomotion have been observed in a wide range of species such as fruit flies [44–46], mice [31],

and humans [47–49]. This makes decreased locomotor activity a particularly consistent aging biomarker that can facilitate comparative work [50]. In zebrafish, age-related decreases in locomotor activity are associated with reduced endurance and swim performance, likely due to sarcopenia (i.e., the loss of muscle mass and strength) [5, 51]. In humans, sarcopenia is also thought to be a key driver of age-related reductions in locomotor activity [52, 53]. Given the overlap in the structure and development of skeletal muscle in zebrafish and mammals [51, 54], our findings lend further support to the use of zebrafish, especially males, as a model system for studying the effects of aging on muscle function and performance.

Aging is thought to be caused by the accumulation of damage to cells over time due to many processes such as telomere shortening and cellular senescence [55]. Telomeres are structures found on the end of chromosomes that protect DNA from degradation and tend to shorten over time [56, 57]. Zebrafish, like mammals, suffer from age-related reductions in telomere length that varies across tissue types [58, 59]. Of interest to the present study, age-related telomere shortening occurs in both the brain and muscles of zebrafish [15, 59]. Telomere shortening is thought to be a key driver of cellular senescence [60], a cellular state marked by loss of proliferative potential and functional changes that contribute to tissue dysfunction [61]. It may be the case that the changes in behavior and locomotor activity we observed are driven by telomere shortening in the brain and muscles. Some of the sex differences in aging may also be explained by telomere shortening. Females of many animals (e.g., humans, reptiles and medaka fish), have longer telomeres than males [62–64]. In zebrafish, no sex differences in telomere length was observed in the brain or heart [15]. However, given that telomere length varies considerably across tissues [59] and between regions and cell types in the brain [65, 66], it may be the case that that sex differences in telomere length are present in skeletal muscle or vary across brain regions to drive some of the age-related sexual dimorphisms observed in the present study. Unraveling the potential contributions of changes in telomeres to the behavioral effects of aging would be a fruitful area for future research.

We did not observe any clear effects of age on either short-term or long-term habituation memory in our study. This is in contrast to prior work that found older zebrafish had deficits in associative learning [18, 20]. This discrepancy may be due to the use of different ages. Prior work used animals ranging in age from 1 to 3 years of age whereas in the present study animals ranged in age from 6 to 12 months. It may be the case that the age difference needs to be larger to see changes in learning. Alternatively, it may be that associative learning is more sensitive to the effects of age in zebrafish than the non-associative habituation learning examined in the present study.

As zebrafish grow in prominence as a model organism in behavioral neurobiology [13, 14] it is critical to understand how biological factors influence their behavior. This is essential for not only identifying what aspects of human behavior can be modeled, but also for deciding what age, sex, and strain of animal may be best suited to address a specific question. The findings of the present study, that a 6 month difference in age can result in behavioral changes, suggests zebrafish may prove to be a useful model organism for studying the interplay of sex and genetics with aging.

## Methods

### Subjects

Animals were female or male TL or TU zebrafish either 6 months post-fertilization (mpf; 24–32 weeks of age) or 12 mpf (52–58 weeks of age). The number of fish per group are as follows with the same number of male and female fish per group: TLs, 6mpf: n's = 16; TLs, 12mpf: n's = 12, TU, 6mpf: n's = 14, TU, 12mpf: n's = 10. All animals were bred and raised at Wayne

State University and were within two generations from fish obtained from the Zebrafish International Resource Center at the University of Oregon. Animals were housed on high density racks using standard conditions (temperature: 27.5 ± 0.5 °C, salinity: 500 ± 10 μS, pH: 7.4 ± 0.2, hardness: 100 mg / L, dissolved oxygen: 97%) and a 14:10 light:dark cycle (lights on at 8:00AM). Water was sourced from the city of Detroit and subject to carbon filtration and water softening prior to reverse osmosis filtration (Evoqua Water Technologies, Troy, MI, USA). Salinity and pH were maintained by the addition of sea salts (Instant Ocean, Blacksburg, VA, USA) and sodium bicarbonate (Aquatic Solutions, Des Moines, IA, USA), respectively. Feeding was twice daily with a dry feed (Gemma 300, Skretting, Westbrook, ME, USA) in the morning and brine shrimp (*Artemia salina*, Brine Shrimp Direct, Ogden, UT, USA) in the afternoon.

Fish were sexed using three secondary sex characteristics: color, shape, and presence of pectoral fin tubercles in males [67]. After experiments, animals were euthanized via immersion in cold water. All procedures were approved by the Wayne State University Institutional Animal Care and Use Committee.

### Novel tank test

At least one week prior to behavioral testing, fish were dual housed as female/male pairs. Two-liter tanks were divided in half using a transparent barrier and pairs of fish were placed in each section of the tank. One hour prior to testing in the novel tank, fish were removed from the racks and allowed to habituate in the procedure room. Fish were then placed individually into the novel tank for six minutes. Water in the testing tank was changed between animals. Following testing, fish remained in the procedure room for an hour prior to being returned to their racks.

The novel tank test was performed in five-sided tanks (15 ˣ 15 ˣ 15 cm) made from frosted acrylic (TAP Plastics, Stockton, CA, USA) and filled with 2.5 L of fish facility water (to a height of 12 cm). Tanks were open from above and placed in an enclosure of white plasticore to prevent the influence of external stimuli and to diffuse light. Videos were recorded using D435 Intel RealSense™ depth-sensing cameras (Intel, Santa Clara, CA, USA) connected to Linux workstations. Cameras were mounted 20 cm above the tanks.

### Animal tracking and behavior

Color videos were extracted from files generated by the Intel RealSense™ D435 cameras and five points along the fish were tracked using DeepLabCut [68]. Using custom written Python code, tracking was overlaid with the depth video from the cameras to generate three-dimensional swim traces for each fish. Four behavioral parameters were extracted as previously described [37]: (1) distance of fish from the bottom of the tank was measured by fitting a plane to the tank bottom and calculating the distance between the plane and the fish, (2) center distance was measured by calculating the distance of the fish from an imaginary line down the center of the tank and the fish, (3) distance travelled was calculated using Euclidean distance between points, and (4) percent of the tank explored was calculated by dividing the tank into one thousand equally spaced voxels and determining the proportion of voxels visited. Prior to calculations, swim traces were smoothed using a Savitzky-Golay filter with an order of three and a length of seven frames [69].

### Statistical analysis

Analysis was done using R version 4.3.0 [70] and visualized using ggplot2 [71]. Data were analyzed for significance using 2 ˣ 2 (age ˣ sex) ANOVAs as appropriate. Significant effects were

followed up with independent samples t-tests within sex and corrected for multiple comparisons using the false discovery rate [72]. When assessing how behavior changed over time, we used one-sample t-tests ($\mu = 0$) and corrected for multiple comparisons as above. Effect sizes are reported as $\eta^2$ for ANOVAs and Cohen's d for t-tests and interpreted as small ($0.01 < \eta^2 < 0.06$; $0.2 < d < 0.5$), medium ($0.06 \leq \eta^2 < 0.14$; $0.5 \leq d < 0.8$) or large ($\eta^2 \geq 0.14$; $d \geq 0.8$) based on [73].

## Supporting information

**S1 Data.**
(CSV)

## Author Contributions

**Conceptualization:** Justin W. Kenney.

**Data curation:** Justin W. Kenney.

**Formal analysis:** Justin W. Kenney.

**Funding acquisition:** Justin W. Kenney.

**Investigation:** Jacob Hudock.

**Methodology:** Jacob Hudock.

**Project administration:** Justin W. Kenney.

**Resources:** Justin W. Kenney.

**Software:** Justin W. Kenney.

**Supervision:** Justin W. Kenney.

**Visualization:** Justin W. Kenney.

**Writing – original draft:** Justin W. Kenney.

**Writing – review & editing:** Jacob Hudock, Justin W. Kenney.

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
