## [Decision Letter · Decision Letter 0]

2 Jan 2024

PONE-D-23-37201Aging in zebrafish is associated with reduced locomotor activity and strain-dependent changes in predator avoidance behaviors related to anxiety.PLOS ONE

Dear Dr. Kenney,

Thank you for submitting your manuscript to PLOS ONE. After careful consideration, we feel that it has merit but does not fully meet PLOS ONE’s publication criteria as it currently stands. Therefore, we invite you to submit a revised version of the manuscript that addresses the points raised during the review process.

**Two experts in the field have carefully reviewed the manuscript entitled, "Aging in zebrafish is associated with reduced locomotor activity and strain-dependent changes in predator avoidance behaviors related to anxiety."　Their comments are appended below.**

**The first reviewer acknowledged the manuscript is fairly well written, however, the second one points out several serious critiques which preclude its acceptance as it stands. This Academic Editor strongly suggests that the authors would rewrite according to the comments and looks forwards to receiving the necessary replies to each concerns and the revised manuscript.**

We look forward to receiving your revised manuscript.

Kind regards,

Manabu Sakakibara, Ph.D.

Academic Editor

PLOS ONE

Journal Requirements:

3. Did you know that depositing data in a repository is associated with up to a 25% citation advantage (https://doi.org/10.1371/journal.pone.0230416)? If you’ve not already done so, consider depositing your raw data in a repository to ensure your work is read, appreciated and cited by the largest possible audience. You’ll also earn an Accessible Data icon on your published paper if you deposit your data in any participating repository (https://plos.org/open-science/open-data/#accessible-data).

4. To comply with PLOS ONE submissions requirements, in your Methods section, please provide additional information regarding the experiments involving animals and ensure you have included details on methods of euthanasia.

NIGMS R35GM142566

This work was funded by the National Institutes of Health (R35GM142566) to J.W.K

NIGMS R35GM142566

Reviewers' comments:

Reviewer's Responses to Questions

**Comments to the Author**

1. Is the manuscript technically sound, and do the data support the conclusions?

Reviewer #1: Yes

Reviewer #2: Yes

2. Has the statistical analysis been performed appropriately and rigorously? 

Reviewer #1: Yes

Reviewer #2: Yes

3. Have the authors made all data underlying the findings in their manuscript fully available?

Reviewer #1: Yes

Reviewer #2: Yes

4. Is the manuscript presented in an intelligible fashion and written in standard English?

Reviewer #1: Yes

Reviewer #2: Yes

5. Review Comments to the Author

Reviewer #1: This paper presents a highly intriguing study and unveils some fascinating findings. The authors investigated on zebrafish behaviors in the novel tank test, with a specific focus on the impact of age and fish strain. The results revealed that even slight differences in age lead to reduced locomotor activity, while strain-dependent variations were observed in predator avoidance behaviors associated with anxiety. Older TL fish exhibited increased bottom dwelling tendencies, whereas older TU fish displayed decreased thigmotaxis. Overall, this paper was well organized and written. I have only several minor comments as below.

1. I am uncertain about PLOS ONE's specific requirements regarding manuscript line numbers. However, it is common for journals to request authors to include line numbers in their submissions as this can aid the review process.

2. The sentences, such as "The distance from the bottom of the tank is considered a predator avoidance response, which has been extensively investigated in zebrafish (Gerlai, 2010; Kalueff et al., 2013)," and "This behavior has also been interpreted as 'anxiety-like,' where spending more time near the bottom of the tank indicates higher levels of anxiety (Maximino et al., 2010) (page 4, first paragraph in the results section)," were not actually presented as interpretations of the results. Therefore, I would suggest that these sentences be included in the discussion section.

3. “Thigmotaxis (i.e. time spent near the periphery) of a tank is also commonly interpreted as a predator avoidance response in zebrafish, although this interpretation has recently come under question”. What is the problem with use of thigmotaxis as an indicator for predator avoidance response? It is necessary to clarify why this approach is still employed in this study despite its questionable reliability.

Reviewer #2: Behavioral changes induced by aging is an interesting topic, and the specific associations remain unexplored. It is essential that the authors systematically assessed a wide range of behavioral changes in zebrafish of different strains, sexes, and ages, by using zebrafish as a model organism. However, in my opinion, this work lacks a deep investigation of the molecular mechanisms involved. Therefore, I suggest that this manuscript needs major revisions before it can be accepted for publication. Specific comments follow:

1. Continuous line numbers in manuscripts are necessary.

2. Abstract. The fact that the zebrafish may be a useful model organism does not seem to be attributable to the fact that, like humans, it also ages, so please rewrite.

3. Please add the information on behavioral changes in zebrafish of different sexes in the Abstract.

4. The significance labeling in all the figures is not explicit enough, please replace the significance labeling with a clearer one.

5. In general, the authors only compare phenotypic behavioral differences between zebrafish of different ages, sexes, and strains, and I'm more interested in the specific mechanisms involved.

6. Neurobehavioral changes are a hot topic today, but the publications cited throughout the manuscript are too old, so please update the citations.

7. Information on the level of dissolved oxygen and hardness in water is necessary.

8. The author mentions that the sex of the fish is finally determined on the basis of the presence or absence of eggs after dissection, which in my opinion is misleading to the reader. Please delete it.

9. The authors are requested to provide a detailed description of how each behavioral indicator is assessed.

10. The authors should provide details on the origin of the water used for the behavioral tests, as changes in water quality can seriously affect the locomotor behavior of zebrafish.

11. Overall, this work systematically evaluates behavioral indicators in zebrafish of different strains, sexes, and ages. However, so far, it seems to me that this work does not highlight the need for zebrafish as a test subject. In other words, zebrafish are widely used by scholars because of their well-defined and highly homologous genome and conserved disease signaling pathways, but this work does not explore in depth the molecular mechanisms involved in the behavioral changes. I think this section will be of interest to future readers.

12. The discussion section lacks depth, it is more of a summary of the published paper, please rewrite.

6. PLOS authors have the option to publish the peer review history of their article (what does this mean?). If published, this will include your full peer review and any attached files.

Reviewer #1: No

Reviewer #2: No

---

## [Author Response · Author response to Decision Letter 0]

12 Feb 2024

Dear Dr. Manabu Sakakibara,

 Thank you for taking the time to serve as editor for our recently submitted manuscript, “Aging in zebrafish is associated with reduced locomotor activity and strain-dependent changes in predator avoidance behaviors related to anxiety” at PLoS One. We were pleased to learn that the reviewers found our work “highly intriguing”, “well organized”, and an “interesting topic”. 

Below, we have described how we addressed reviewer comments. Comments are in bold with our responses in italics. In the modified manuscript file, changes made are in red. I would also like to note that we made a modest change to the title of the manuscript so that it is more descriptive. It is now: 

“Aging in zebrafish is associated with reduced locomotor activity and strain-dependent changes in bottom dwelling and thigmotaxis.”

Reviewer #1

I am uncertain about PLOS ONE's specific requirements regarding manuscript line numbers. However, it is common for journals to request authors to include line numbers in their submissions as this can aid the review process.

We apologize for the oversight of not including line numbers. They have now been included.

The sentences, such as “The distance from the bottom of the tank is considered a predator avoidance response, which has been extensively investigated in zebrafish (Gerlai, 2010; Kalueff et al., 2013),” and “This behavior has also been interpreted as ‘anxiety-like,’ where spending more time near the bottom of the tank indicates higher levels of anxiety (Maximino et al., 2010) (page 4, first paragraph in the results section),” were not actually presented as interpretations of the results. Therefore, I would suggest that these sentences be included in the discussion section.

We discuss the interpretation of our data in the context of anxiety in the second paragraph of the discussion (lines 309-322). 

“Thigmotaxis (i.e. time spent near the periphery) of a tank is also commonly interpreted as a predator avoidance response in zebrafish, although this interpretation has recently come under question”. What is the problem with use of thigmotaxis as an indicator for predator avoidance response? It is necessary to clarify why this approach is still employed in this study despite its questionable reliability.

We expand on this point in more depth in the discussion (lines 323-345). After this sentence, we have now added a parenthetical reference to the discussion (line 127).

Briefly, in the third paragraph of the discussion we lay out the challenges with the interpretation of thigmotaxis. The issue is that thigmotaxis does not correlate well with other more well-established markers of predator avoidance or anxiety (e.g., freezing or bottom dwelling). Thigmotaxis also does not change within session as would be expected of an anxiety-like/predator avoidance behavior (and how bottom dwelling often does). At the end of the paragraph, we also discuss a novel interpretation of the thigmotaxis data, i.e., that genetic background, perhaps through the natural history of the strain, may mean that some strains of fish express predator avoidance through thigmotaxis, but others do not (lines 339-345). 

Reviewer #2

Continuous line numbers in manuscripts are necessary.

This has now been included. We apologize for the oversight. 

Abstract. The fact that the zebrafish may be a useful model organism does not seem to be attributable to the fact that, like humans, it also ages, so please rewrite.

The abstract indicates that zebrafish may be a useful model for human aging because they age gradually, not simply because they age. This contrasts with the very slow to negligible aging in other animals of indeterminate growth (e.g. many reptiles and sea urchins). To make this point more prominent, we have added a couple of lines to the beginning of the introduction (lines 57-59). 

Please add the information on behavioral changes in zebrafish of different sexes in the Abstract.

This has now been included in the abstract 

The significance labeling in all the figures is not explicit enough, please replace the significance labeling with a clearer one.

We agree with the reviewer that the significance labeling for the results of the ANOVAs was not particularly clear. To improve the labelling, we now only refer to the ANOVA results in the text, and the figures only include the results of post-hoc tests within sex that we performed after significant ANOVAs or the one-sample t-tests. We have also updated the text accordingly. We have also updated the conclusions and abstract to match the more refined results of the post-hoc tests. We appreciate the suggestion as it has enhanced the readability and interpretation of the results. 

In general, the authors only compare phenotypic behavioral differences between zebrafish of different ages, sexes, and strains, and I'm more interested in the specific mechanisms involved.

Identifying the underlying mechanisms are of interest to us as well. However, we do not have the resources at the moment to do this research. Nonetheless, we believe our behavioral findings are worth reporting as they may spur others to use zebrafish as a model to uncover the molecular mechanisms that underlie the effects of aging on behavior.

Neurobehavioral changes are a hot topic today, but the publications cited throughout the manuscript are too old, so please update the citations.

We have added some updated citations (lines 314-315, 318, 348, 351). Nonetheless, we kept many of the older citations because we believe it is important to give credit to those who first made a discovery.

Information on the level of dissolved oxygen and hardness in water is necessary.

This has now been added to the methods section (lines 429-430).

The author mentions that the sex of the fish is finally determined on the basis of the presence or absence of eggs after dissection, which in my opinion is misleading to the reader. Please delete it.

This line has now been removed.

The authors are requested to provide a detailed description of how each behavioral indicator is assessed.

This has now been included on lines 457-469

The authors should provide details on the origin of the water used for the behavioral tests, as changes in water quality can seriously affect the locomotor behavior of zebrafish.

We have now included this in the methods section (lines 431-434). Briefly, water from the city of Detroit was filtered through a carbon filter and water softener before being reverse osmosis filtered. Sea salts and sodium bicarbonate are then added to the water to maintain the pH and salinity at the desired levels. 

Overall, this work systematically evaluates behavioral indicators in zebrafish of different strains, sexes, and ages. However, so far, it seems to me that this work does not highlight the need for zebrafish as a test subject. In other words, zebrafish are widely used by scholars because of their well-defined and highly homologous genome and conserved disease signaling pathways, but this work does not explore in depth the molecular mechanisms involved in the behavioral changes. I think this section will be of interest to future readers.

The discussion section lacks depth, it is more of a summary of the published paper, please rewrite.

In our discussion we try not to speculate too much beyond what our data supports. However, we do offer some novel ideas, such as the discussion on thigmotaxis (lines 323-345) where we suggest that whether thigmotaxis indicates an anxiety-like state in zebrafish may depend on the background strain of the animals. We also suggest that sarcopenia (i.e., muscle loss) may contribute to the changes in locomotor activity we observe (lines 354-362). Nonetheless, we added a paragraph on the potential role that telomere biology may play in the interplay between sex and aging we observed (lines 362-383). 

Sincerely,

Justin W. Kenney & Jacob Hudock

---

## [Decision Letter · Decision Letter 1]

26 Feb 2024

Aging in zebrafish is associated with reduced locomotor activity and strain dependent changes in bottom dwelling and thigmotaxis

PONE-D-23-37201R1

Dear Dr. Kenney,

We’re pleased to inform you that your manuscript has been judged scientifically suitable for publication and will be formally accepted for publication once it meets all outstanding technical requirements.

Kind regards,

Manabu Sakakibara, Ph.D.

Academic Editor

PLOS ONE

Additional Editor Comments (optional):

Reviewers' comments:

Reviewer's Responses to Questions

**Comments to the Author**

1. If the authors have adequately addressed your comments raised in a previous round of review and you feel that this manuscript is now acceptable for publication, you may indicate that here to bypass the “Comments to the Author” section, enter your conflict of interest statement in the “Confidential to Editor” section, and submit your "Accept" recommendation.

Reviewer #1: All comments have been addressed

2. Is the manuscript technically sound, and do the data support the conclusions?

Reviewer #1: Yes

3. Has the statistical analysis been performed appropriately and rigorously? 

Reviewer #1: Yes

4. Have the authors made all data underlying the findings in their manuscript fully available?

Reviewer #1: Yes

5. Is the manuscript presented in an intelligible fashion and written in standard English?

Reviewer #1: Yes

6. Review Comments to the Author

Reviewer #1: All of the issues I concerned have been well addressed. The manuscript has been improved a lot, and I have no further suggestion.

7. PLOS authors have the option to publish the peer review history of their article (what does this mean?). If published, this will include your full peer review and any attached files.

Reviewer #1: No

---

## [Editor Report · Acceptance letter]

17 Mar 2024

PONE-D-23-37201R1 

PLOS ONE

Dear Dr. Kenney, 

I'm pleased to inform you that your manuscript has been deemed suitable for publication in PLOS ONE. Congratulations! Your manuscript is now being handed over to our production team.

Kind regards, 

on behalf of

Dr. Manabu Sakakibara 

Academic Editor

PLOS ONE